# DeepArchitect: Automatically Designing and Training Deep Architectures

## Abstract

In deep learning, performance is strongly affected by the choice of architecture and hyperparameters. While there has been extensive work on automatic hyperparameter optimization for simple spaces, complex spaces such as the space of deep architectures remain largely unexplored. As a result, the choice of architecture is done manually by the human expert through a slow trial and error process guided mainly by intuition. In this paper we describe a framework for automatically designing and training deep models. We propose an extensible and modular language that allows the human expert to compactly represent complex search spaces over architectures and their hyperparameters. The resulting search spaces are tree-structured and therefore easy to traverse. Models can be automatically compiled to computational graphs once values for all hyperparameters have been chosen. We can leverage the structure of the search space to introduce different model search algorithms, such as random search, Monte Carlo tree search (MCTS), and sequential model-based optimization (SMBO). We present experiments comparing the different algorithms on CIFAR-10 and show that MCTS and SMBO outperform random search. We also present experiments on MNIST, showing that the same search space achieves near state-of-the-art performance with a few samples. These experiments show that our framework can be used effectively for model discovery, as it is possible to describe expressive search spaces and discover competitive models without much effort from the human expert. Code for our framework and experiments has been made publicly available.

## 1 Introduction

Deep learning has seen a surge in popularity due to breakthroughs in applications such as computer vision, natural language processing, and reinforcement learning (He et al., 2016; Karpathy & Fei-Fei, 2015; Silver et al., 2016; Sutskever et al., 2014). An important observation in much of the recent work is that complex architectures are important for achieving high performance (He et al., 2016; Mnih et al., 2013). Larger datasets and more powerful computing infrastructures are likely to increase our ability to effectively train larger, deeper, and more complex architectures. However, improving the performance of a neural network is not as simple as adding more layers or parameters—it often requires clever ideas such as creating more branches (Szegedy et al., 2015) or adding skip connections (He et al., 2016). Even popular techniques such as dropout (Srivastava et al., 2014) and batch normalization (Ioffe & Szegedy, 2015) do not always lead to better performance, and need to be judiciously applied to be helpful.

Currently, choosing appropriate values for these architectural hyperparameters requires close supervision by a human expert, in a trial and error manual search process largely guided by intuition. The expert is burdened by having to make the large number of choices involved in the specification of a deep model. Choices interact in non-obvious ways and strongly impact performance. The typical workflow has the expert specify a single model, train it, and compute a validation score. Based on the validation score, previous experience, and information gathered during training, the expert decides if the trained model is satisfactory or not. If the model is considered unsatisfactory, the expert has to think about model variations that may lead to better performance.

From the perspective of the expert, it would be convenient to search over architectures automatically, just as we search over simple scalar hyperparameters, such as the learning rate and the regularization

coefficient. Ideally, the expert would have control in setting up the search space to incorporate inductive biases about the task being solved and constraints about computational resources. Prior to this work, achieving this goal was hard because expressing model search spaces using general hyperparameter optimization tools requires the human expert to manually distill a set of relevant scalar architectural hyperparameters.

The main contributions of our work are

1. a modular, compositional, and extensible language for compactly representing expressive search spaces over models that
   (a) gives control to the human expert over what model variations to consider;
   (b) makes it easy to automatically search for performant models in the search space;
   (c) allows models to be directly compiled to computational graphs without the human expert having to write additional code.
2. model search algorithms that rely on the tree-structured search spaces induced by our language to systematically and efficiently search for performant models; namely, we
   (a) show that by using constructs in our language, even random search can be effective;
   (b) compare different model search algorithms experimentally, and show that random search is outperformed by algorithms that leverage the structure of the search space to generalize more effectively across different models.

The main differences between our work and previous work are that we develop a modular, composable and extensible language, focusing on the problem of searching over deep architectures. This focus allows the expert to compactly set up a search space, search over it, and automatically compile models to their corresponding computational graphs. Our language can be seen as an effort to combine the functionalities of a deep model specification language (e.g., Tensorflow (Abadi et al., 2016)) and a structured hyperparameter search language (e.g., Hyperopt (Yamins et al., 2013)).

## 2 RELATED WORK

Model search has a long and rich history in machine learning and statistics. There has been a wide variety of theoretical and empirical research in this area (Agarwal et al., 2011; Bergstra et al., 2011; Bergstra & Bengio, 2012; Sabharwal et al., 2015), including Bayesian optimization methods (Hutter et al., 2011; Kandasamy et al., 2015; Snoek et al., 2012). However, conventional methods are primarily designed for searching over hyperparameters living in Euclidean space. Such methods are ill suited in today's context, where the discrete architectural choices are just as important as the numerical values of the hyperparameters. Searching over architectures using current hyperparameter optimization algorithms requires the expert to distill structural choices into scalar hyperparameters. As a result, typically only a few simple global structural hyperparameters are considered, e.g., the depth of the network or whether to use dropout or not. This constrains the richness of the search space, preventing the expert from finding unexpected model variations leading to better performance; e.g., perhaps dropout is useful only after certain types of layers, or batch normalization only helps in the first half of the network.

Architecture search has also been considered under the topic of neuroevolution (Stanley & Miikkulainen, 2002), which uses evolutionary (i.e., genetic) strategies to define and search a space of models. In classical approaches, neuroevolution attempts to jointly choose the topology and the parameters of the architecture using genetic algorithms.

Architecture search has received renewed interest recently. Wierstra et al. (2005), Floreano et al. (2008), and Real et al. (2017) use evolutionary algorithms which start from an initial model and evolve it based on its validation performance. Zoph & Le (2017) propose a reinforcement learning procedure based on policy gradient for searching for convolutional and LSTM architectures. Baker et al. (2016) propose a reinforcement learning procedure based on Q-learning for searching for convolutional architectures.

Unfortunately all these approaches consider fixed hard-coded model search spaces that do not easily allow the human expert to incorporate inductive biases about the task being solved, making them unsuitable as general tools for architecture search. For example, evolutionary approaches require

an encoding for the models in the search space and genetic operators (e.g., mutation and crossover) which generate encodings for new models out of encodings of old ones. These aspects are hand-crafted and hard-coded so it is hard for the human expert to change the search space in flexible ways. Perhaps different model encodings or genetic operators can be considered, but these knobs give somewhat loose and indirect control over the model search space. The reinforcement learning approaches considered suffer from similar issues—the search spaces are hard-coded and not easily modifiable. None of these approaches have the compositionality, modularity, and extensibility properties of our language.

Bergstra et al. (2011) propose Tree of Parzen Estimators (TPE), which can be used to search over structured hyperparameter spaces, and use it to tune the hyperparameters of a Deep Boltzmann Machine. Yamins et al. (2013) use TPE to search for values of the hyperparameters of a computer vision system, and show that it can find better values than the best ones previously known.

TPE is a general hyperparameter search algorithm, and therefore requires considerable effort to use—for any fixed model search space, using TPE requires the human expert to distill the hyperparameters of the search space, express the search space in Hyperopt (Yamins et al., 2013) (an implementation of TPE), and write the code describing how values of the hyperparameters in the search space compile to a computational graph. In contrast, our language is modular and composable in the sense that:

1. search spaces (defined through modules) are constructed compositionally out of simpler search spaces (i.e., simpler modules);

2. hyperparameters for composite modules are derived automatically from the hyperparameters of simpler modules;

3. once values for all hyperparameters of a module have been chosen, the resulting model can be automatically mapped to a computational graph without the human expert having to write additional code.

## 3    ROADMAP TO THE DEEPARCHITECT FRAMEWORK

Our framework reduces the problem of searching over models into three modular components: the *model search space specification language*, the *model search algorithm*, and the *model evaluation algorithm*.

**Model Search Specification Language:**    The model search space specification language is built around the concept of a modular *computational module*. This is akin to the concept of a module (Bottou & Gallinari, 1991) used in deep learning frameworks such as Torch (Collobert et al., 2011): by implementing the module interface, the internal implementation becomes irrelevant. These modules allow one to express easily complex design choices such as whether to include a module or not, choose between modules of different types, or choose how many times to repeat a module structure. The main insight is that complex modules can be created *compositionally* out of simpler ones. The behavior of complex modules is generated automatically out of the behavior of simpler modules. Furthermore, our language is extensible, allowing the implementation of new types of modules by implementing a high-level interface local to the module.

**Model Search Algorithm:**    The way the model search space is explored is determined by the *model search algorithm*. This part of the framework decides how much effort to allocate to each part of the search space based on the performance observed for previous models. The model search algorithm typically requires a *model evaluation algorithm* that computes the performance of a fully specified model. The search algorithm will then use this information to determine which models to try next. The search algorithm interacts with the search space only through a minimal interface that allows it to traverse the space of models and evaluate models discovered this way. This interface is the same irrespective of the specific search space under consideration. We experiment with different search algorithms, such as Monte Carlo tree search (Browne et al., 2012) and Sequential Model Based Optimization (Hutter et al., 2011).

**Model Evaluation Algorithm:** Having fully specified a model, i.e., having reached a leaf in the tree defined by our model search space, we can evaluate how good this model is according to some criterion defined by the expert. This typically involves training the model on a training set and evaluating it on a validation set. The training procedure often has multiple hyperparameters that can be tuned (e.g., the choice of the optimization algorithm and its hyperparameters, and the learning rate schedule). If the expert does not know how to write down a reasonable training procedure for every model in the search space, the expert can introduce hyperparameters for the evaluation algorithm and search over them using our specification language.

Any of the above components can be changed, improved, or extended, while keeping the others fixed. The fact that different components interact only through well-defined interfaces makes it possible to extend and reuse this framework. We believe that DeepArchitect will be an interesting platform for future research in deep learning and hyperparameter tuning for architecture search.

## 4 MODEL SEARCH SPACE SPECIFICATION LANGUAGE

### 4.1 SEARCH SPACE DEFINITION

The *computational module* is the fundamental unit of our model search space specification language. We define a computational module as a function

$$f : n \to (\mathcal{H} \to (\mathbb{R}^p \to (\mathbb{R}^n \to \mathbb{R}^m))), \tag{1}$$

where $n$ is the dimensionality of the *input*, $\mathcal{H}$ is the set of valid values for the *hyperparameters*, $p$ is the number of *parameters*, and $m$ is the dimensionality of the *output*. The set $\mathcal{H}$ can be structured or simply the cross product of scalar hyperparameter sets, i.e., $\mathcal{H} = \mathcal{H}_1 \times \ldots \times \mathcal{H}_H$, where $H$ is the number of scalar hyperparameters. The set $\mathcal{H}$ is assumed to be discrete in both cases.

Definition (1) merits some discussion. For conciseness we have not explicitly represented it, but *the number of parameters $p$ and the output dimensionality $m$ can both be functions of the input dimensionality $n$ and the chosen hyperparameter values $h \in \mathcal{H}$.* For example, an affine module with $h$ dense hidden units has output dimensionality $m = h$ and number of parameters $p = (n + 1)h$: a weight matrix $W \in \mathbb{R}^{h \times n}$ and a bias vector $b \in \mathbb{R}^h$. A similar reasoning can be carried out for a convolutional module: the number of parameters $p$ depends on the input dimensionality, the number of filters, and the size of the filters; the dimensionality of the output $m$ depends on the input dimensionality, the number of filters, the size of the filters, the stride, and the padding scheme. The fact that $p$ and $m$ are functions of the input dimensionality and the chosen hyperparameter values is one of the main observations that allows us to do architecture search—*once we know the input dimensionality and have fixed values for the hyperparameters, the structure of the computation performed by the module is determined*, and this information can be propagated to other modules. We say that a module is *fully specified* when values for all hyperparameters of the module have been chosen and the input dimensionality is known.

We focus on search spaces for architectures that have a *single input terminal* and a *single output terminal*. By this, we only mean that the input and output of the module have to be a *single* tensor of arbitrary order and dimensionality. For example, convolutional modules take as input an order three tensor and return as output an order three tensor, therefore they are single-input single-output modules under our definition. We also assume that the output of a module is used as input to at most a single module, i.e., we *assume no output sharing*.

These restrictions were introduced to simplify exposition. The single-input single-output case with no sharing is simpler to develop and exemplifies the main ideas that allow us to develop a framework for automatic architecture search. The ideas developed in this work extend naturally to the multiple-input multiple-output case with sharing. Additionally, often we can represent modules that are not single-input single-output by defining new modules that encapsulate many signal paths from input to output. For example, a residual module (He et al., 2016) can be treated in our framework by noting that it is single-input before the skip connection split and single-output after the skip connection merge. Many top performing architectures, such as AlexNet (Krizhevsky et al., 2012), VGG (Simonyan & Zisserman, 2014), and ResNet (He et al., 2016), are captured in our language.

We distinguish between *basic computational modules* and *composite computational modules*. Basic modules do some well defined transformation. Affine, batch normalization, and dropout are ex-

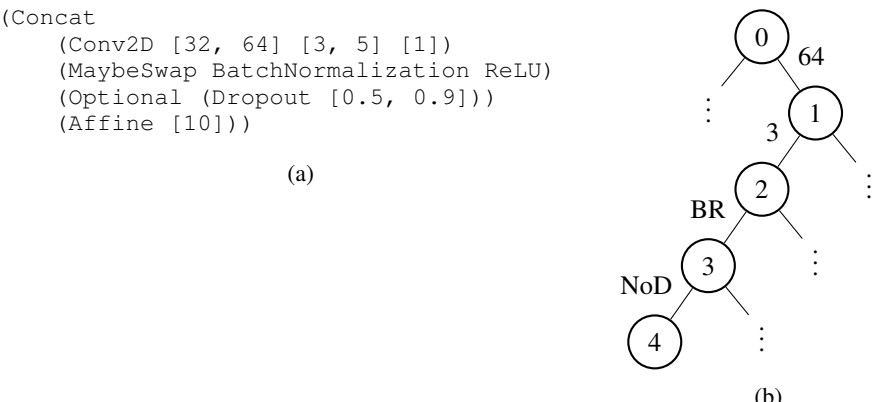

```
(Concat
    (Conv2D [32, 64] [3, 5] [1])
    (MaybeSwap BatchNormalization ReLU)
    (Optional (Dropout [0.5, 0.9]))
    (Affine [10]))
```

(a)

(b)

Figure 1: (a) A simple search space with 24 different models. (b) A path through the search space encoding a convolutional module with 64 filters of size $3 \times 3$, with stride 1, followed by batch normalization, ReLU and affine modules. The model does not use dropout. Branches encoding hyperparameters with a single choice were omitted.

amples of basic modules. Composite modules are defined in terms of other (composite or basic) modules, i.e., the instantiation of a composite module takes other modules as arguments. Composite modules may introduce hyperparameters of their own and inherit hyperparameters of the modules taken as arguments. For example, an `Or` module takes a list of modules and chooses one of the modules to use. It introduces a discrete hyperparameter for which module to use, and chooses values for the hyperparameters of the chosen module; the hyperparameters available are conditional on the choice of the module to use. Most of the representational power of our language arises from the compositionality of composite and basic modules.

The ideas developed in this section are perhaps best illustrated with an example. See Figure 1a for the definition of an example search space in LISP-like pseudocode that closely parallels our implementation. The search space, which results from the composition of several modules, and therefore is also a module itself, encodes 24 different models, corresponding to the different 24 possible paths from the root to the leaves of the tree. The space is defined using three composite modules (`Concat`, `MaybeSwap`, and `Optional`) and five basic modules (`Conv2D`, `BatchNormalization`, `ReLU`, `Dropout`, and `Affine`). `Concat` introduces no additional hyperparameters, but it has to specify all the modules that have been delegated to it; `MaybeSwap` introduces a binary hyperparameter that encodes whether to swap the order of the pair of modules or not; `Optional` introduces a binary hyperparameter that encodes whether to include the module or not. The behavior of the basic modules in Figure 1a is simple: `Conv2D` takes lists of possible values for the number of filters, the size of the filters, and the stride; `BatchNormalization` and `ReLU` have no hyperparameters; `Dropout` takes a list for the possible values for the dropout probability; `Affine` takes a list for the possible values of the number of hidden units.

Choosing different values for the hyperparameters of the composite modules may affect the structure of the resulting architecture, while choosing different values for the hyperparameters of the basic modules only affects the structure of the corresponding local transformations. The search space of Figure 1a results from the composition of basic and composite modules; therefore it is a module itself and can be characterized by its input, output, parameters, and hyperparameters. Our set of composite modules in not minimal: e.g., given an `Empty` basic module, which has no hyperparameters or parameters and simply does the identity transformation, and a `Or` composite module, which introduces an extra hyperparameter encoding the choice of a specific module in its list, the composite modules `Optional` and `MaybeSwap` can be defined as `(Optional B) = (Or Empty B)` and `(MaybeSwap B1 B2) = (Or (Concat B1 B2), (Concat B2 B1))`.

## 4.2 SEARCH SPACE TRAVERSAL

Given a search space defined by a module, there is an underlying tree over fully specified models: we build this tree by sequentially assigning values to each of the hyperparameters of the module.

Each internal node in the tree corresponds to some partial assignment to the hyperparameters of the module, and each terminal node (i.e., each leaf) corresponds to a fully specified model. We can also think about an internal node as corresponding to the state of a module before assigning a value to the next unassigned hyperparameter. The branching factor of a node corresponds to the number of possible values for the hyperparameter under consideration at that node, and traversing a specific edge from that node to a child corresponds to assigning the value encoded by that edge to the hyperparameter under consideration. As a tree has a single path between the root and any leaf, the paths from root to leaves are in one-to-one correspondence with fully specified models. A leaf is reached when there are no hyperparameters left to specify.

In Figure 1b we have drawn a path through the search space of Figure 1a from the root (labeled node $0$), where all hyperparameters are unassigned, to a terminal node (labeled node $4$), where all hyperparameters have been assigned values. Each branch in the tree corresponds to the assignment of some value to some hyperparameter. At node $0$, we are choosing between $32$ or $64$ filters; at node $1$, we are choosing between filters of size $3$ or $5$; at node $2$, we are choosing between applying batch normalization before or after ReLU; at node $3$, we are choosing whether to do dropout or not. Node $4$ is terminal and corresponds to a fully specified model. Decisions at each node are conditional on decisions previously made. Internal nodes with a single child (i.e., branches for hyperparameters with a single possible value) have been collapsed and omitted from Figure 1a. Other paths may have different lengths, e.g., picking a path through the right child of node $3$ corresponds to adding a `Dropout` module, which requires an additional hyperparameter choice for the dropout probability when compared to the path from the root to node $4$.

Search spaces arising from module composition have their traversal functionality automatically derived from the traversal functionality of their component modules: a basic module knows how to sequentially assign values to its hyperparameters, and a composite module knows how to sequentially assign values to its hyperparameters and call the sequential assignment functionality for its component modules. This is akin to recursive expression evaluation in programming languages.

To traverse the search space, i.e., to assign values to all hyperparameters of the module defining the search space, all that it is needed is that each module knows how to sequentially specify itself. *Modules resulting from the composition of modules will then be automatically sequentially specifiable*. The three *local* operations that a module needs to implement for traversal are: *to test whether it is fully specified* (i.e., whether it has reached a leaf yet); if it is not specified, *to return which hyperparameter it is specifying and what are the possible values for it*; and given a choice for the current hyperparameter under consideration, *to traverse the edge to the child of the current node corresponding to chosen value*.

### 4.3 COMPILATION

Once values for all hyperparameters of a module have been chosen, the fully specified model can be automatically mapped to its corresponding *computational graph*. We call this mapping *compilation*. This operation only requires that each module knows how to locally map itself to a computational graph: compilation is derived recursively from the compilation of simpler modules. For example, if we know how to compile `Conv2D`, `ReLU`, and `Or` modules, we will automatically be able to compile all modules built from them. This behavior is also similar to recursive expression evaluation in programming languages.

## 5 MODEL SEARCH ALGORITHMS

In this section, we consider different search algorithms that are built on top of the functionality described above. Some of these algorithms rely on the search space being tree structured. One of the challenges of our setting is that deep models are expensive to train, so unless we have access to extraordinary computational resources, only a moderate number of evaluations will be practical.

### 5.1 RANDOM SEARCH

Random search is the simplest algorithm that we can consider. At each node of the tree, we choose an outgoing edge uniformly at random, until we reach a leaf node (i.e., a model). Even just random

search is interesting, as the model search space specification language allows us to capture expressive structural search spaces. Without our language, randomly selecting an interesting architecture to try would not be possible without considerable effort from the human expert.

## 5.2 MONTE CARLO TREE SEARCH

Monte Carlo tree search (MCTS) (Browne et al., 2012; Kocsis & Szepesvári, 2006) is an approximate planning technique that has been used effectively in many domains (Silver et al., 2016). Contrary to random search, MCTS uses the information gathered so far to steer its policy towards better performing parts of the search space. MCTS maintains a search tree that is expanded incrementally one node at a time. MCTS uses two policies: a *tree policy*, which determines the path to be traversed from the root to the frontier of the already expanded tree; and a *rollout policy*, which determines the path to be traversed from the frontier of the already expanded tree until a leaf is reached. Once a leaf is reached, the model encoded by it is evaluated (e.g., trained on the training set and evaluated on the validation set), and the resulting score is used to update the statistics of the nodes in the currently expanded tree in the path to the leaf. Each node in the expanded tree keeps statistics about the number of times it was visited and the average score of the models that were evaluated in the subtree at that node. The rollout policy is often simple, e.g., the random policy described in Section 5.1.

The tree policy typically uses an upper confidence bound (UCB) approach. Let $n$ be the number of visits of a node $v \in \mathcal{T}$, where $\mathcal{T}$ denotes the currently expanded tree, and $n_1, \ldots, n_b$ and $\bar{X}_1, \ldots, \bar{X}_b$ be, respectively, the number of visits and the average scores of the $b$ children of $v$. The tree policy at $x$ chooses to traverse an edge corresponding to a child maximizing the UCB score:

$$\max_{i \in \{1, \ldots, b\}} \bar{X}_i + 2c\sqrt{\frac{2 \log n}{n_i}}, \tag{2}$$

where $c \in \mathbb{R}_+$ is a constant capturing the trade-off between exploration and exploitation—larger values of $c$ correspond to larger amounts of exploration. If at node $x$, some of its children have not been added to the tree, there will be some $i \in \{1, \ldots, b\}$ for which $n_i = 0$; in this case we define the UCB score to be infinite, and therefore, unexpanded children always take precedence over expanded children. If multiple unexpanded children are available, we expand one uniformly at random.

## 5.3 MONTE CARLO TREE SEARCH WITH TREE RESTRUCTURING

When MCTS visits a node in the expanded part of the tree, it has to expand all children of that node before expanding any children of its currently expanded children. This is undesirable when there are hyperparameters that can take a large number of related values.

We often consider hyperparameters which take *numeric* values, and similar values result in similar performance. For example, choosing between $64$ or $80$ filters for a convolutional module might not have a dramatic impact on performance. A way of addressing such hyperparameters is to restructure the branches of the tree by doing bisection. Assume that the set of hyperparameters has a natural ordering. At a node, rather than committing directly to a value of the hyperparameter, we commit sequentially—first we decide if we are choosing a value in the first or second half of the set of hyperparameters, and then we recurse on the chosen half until we have narrow it down to a single value. See an example tree in Figure 2a and the corresponding restructured tree in Figure 2b.

Tree restructuring involves a tradeoff between depth and breadth: the tree in Figure 2a has depth 1, while the tree in Figure 2b has depth 3. The restructured tree can have better properties in the sense that there more sharing between different values of the hyperparameters. We could also consider restructured trees with branching factors different than two, again trading off depth and breadth. If the branching factor of the restructured tree is larger than the number of children of the hyperparameter, the restructuring has no effect, i.e., the original and restructured trees are equal. The restructuring operation allows MCTS to *effectively consider hyperparameters with a large number of possible values*.

## 5.4 SEQUENTIAL MODEL BASED OPTIMIZATION

MCTS is tabular in the sense that it keeps statistics for each node in the tree. While the restructuring operation described in Section 5.3 increases sharing between different hyperparameter values, it still

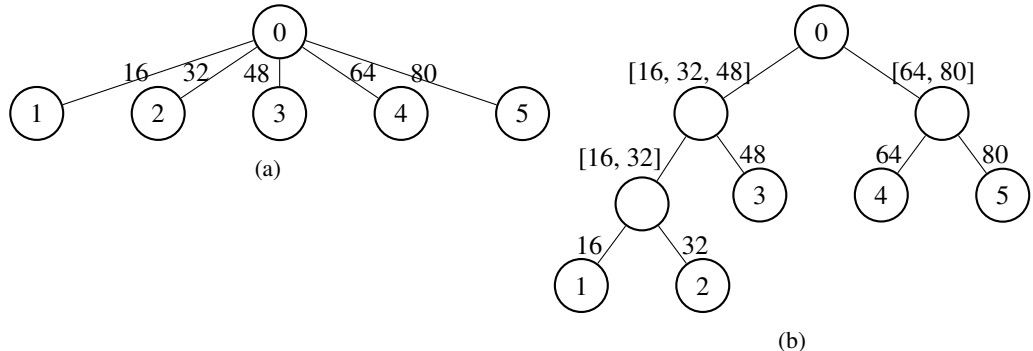

Figure 2: (a) A tree encoding an hyperparameter and its five possible values. MCTS applied to this tree is sample-inefficient as there is no sharing of information between the different child nodes. (b) The result of restructuring the tree with bisection. MCTS applied to this tree results in more sharing when compared to the original tree. For example, sampling a path reaching node 1 provides information about nodes 1, 2, and 3.

suffers from the problem that nodes have no way of sharing information other than through common ancestors. This is problematic because differences in hyperparameter values at the top levels of the tree lead to little sharing between models, even if the resulting models happen to be very similar.

Sequential Model Based Optimization (SMBO) (Hutter et al., 2011) allows us to address this problem by introducing a surrogate function which can be used to capture relationships between models and how promising it is to evaluate any specific model. The surrogate function can use expressive features to capture architecture patterns that influence performance, e.g., features about sequences of basic modules that occur in the model.

The surrogate function can then be optimized to choose which model to evaluate next. Exactly optimizing the surrogate function over a search space can be difficult as often there is a combinatorially large number of models. To approximately optimize the surrogate function, we do some number of random rollouts from the root of the tree until we hit leaf nodes (i.e., models), we evaluate the surrogate function (i.e., we determine, according to the surrogate function, how promising it is to evaluate that model), and evaluate the model that has the highest score according to the surrogate function. We also introduce an exploratory component where we flip a biased coin and choose between evaluating a random model or evaluating the best model according to the surrogate function. The surrogate function is updated after each evaluation.

In our experiments, we use a simple surrogate function: we train a ridge regressor to predict model performance, using the models evaluated so far and their corresponding performances as training data. We only use features based on $n$-grams of sequences of basic modules, disregarding the values of the hyperparameters. More complex features, surrogate functions, and training losses are likely to lead to better search performance, but we leave these to future work.

## 6 MODEL EVALUATION ALGORITHMS

As a reminder, once we assign values to all hyperparameters of the module defining the search space, we need to compute a score for the resulting model, i.e., a score for the path from the root to the corresponding leaf encoding the model to evaluate. The specific way to compute scores is defined by the human expert, and it typically amounts to training the model on a training set and evaluating the trained model on a validation set. The score of the model is the resulting validation performance. The training process often has its own hyperparameters, such as: what optimization algorithm to use and its corresponding hyperparameters, the learning rate schedule (e.g., the initial learning rate, the learning rate reduction multiplier, and how many epochs without improving the validation performance the algorithm waits before reducing the learning rate), how many epochs without improving the validation performance the algorithm waits before terminating the training process (i.e., early stopping), and what data augmentation strategies to use and their corresponding hyperparameters. The behavior of the evaluation algorithm with respect to the values of its hyperparameters is defined

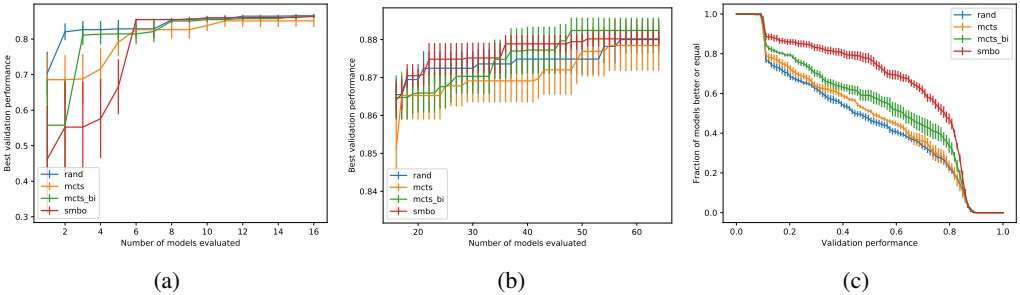

Figure 3: (a, b) Average maximum validation score achieved as a function of the number of evaluation across five repetitions. The error bars indicate standard error. The range of 64 evaluations is split into two plots for clearer visualization. (c) Percentage of models above a given validation threshold performance. MCTS with bisection and SMBO outperform random search. The error bars have size equal to the standard error.

by the expert for the task being considered, so the compilation step described in Section 4.3 for this functionality has to be implemented by the expert. Nonetheless, these *user hyperparameters* can be included in the search space and searched over in the same way as the architecture hyperparameters described in Section 4.1.

## 7 EXPERIMENTS

We illustrate how our framework can be used to search over all hyperparameters of a model, i.e., both architecture and training hyperparameters, using only high-level insights. We choose a search space of deep convolutional models based around the ideas that depth is important, batch normalization helps convergence, and dropout is sometimes helpful. We search over architectures and evaluate our models on CIFAR-10 (Krizhevsky, 2009).

The training hyperparameters that we consider are whether to use ADAM or SGD with momentum, the initial learning rate, the learning rate reduction multiplier, and the rate reduction patience, i.e., how many epochs without improvement to wait before reducing the current learning rate. We use standard data augmentation techniques: we zero pad the CIFAR-10 images to size $40 \times 40 \times 3$, randomly crop a $32 \times 32$ portion, and flip horizontally at random. We could search over these too if desired.

We compare the search algorithms described in Section 5 in terms of the best model found, according to validation performance, as a function of the number of evaluations. We run each algorithm 5 times, for 64 model evaluations each time. All models were trained for 30 minutes on GeForce GTX 970 GPUs in machines with similar specifications.

In Figure 3a and Figure 3b, we see that all search algorithms find performant solutions (around 89% accuracy) after 64 evaluations. In Figure 3a, we see that for fewer than 6 evaluations there is considerable variance between the different algorithms; the more sophisticated model search algorithms are not able to outperform random search with so few evaluations. In Figure 3b, we see that both SMBO and MCTS with bisection eventually outperform random search; MCTS with bisection starts outperforming random search around 32 evaluations, while for SMBO, it happens around 16 evaluations.

Surprisingly, MCTS without restructuring does not outperform random search. We think that this is because there are too many possible values for the first few hyperparameters in the tree, so MCTS will not be able to identify and focus on high-performance regions of the search space within the number of evaluations available. MCTS with bisection and SMBO do not suffer from these problems, and therefore can identify and focus on high performance regions of the search space earlier. In addition to achieving a higher top accuracy, MCTS with bisection and SMBO evaluate a larger fraction of high-performance models when compared to random search, as can be seen in Figure 3c.

The main goal of the previous experiment is to show that more complex model search algorithms can outperform random search by better leveraging the structure of the search. We are not attempting to achieve state-of-the-art performance. We now show that using the same search space on MNIST with a larger time budget leads to close to state-of-the-art performance. The data augmentation scheme is slightly different, as we no longer randomly flip the image horizontally, but now consider random rotations where the maximum angle of rotation is also added as a hyperparameter to the search space.

In this experiment, we randomly sample 16 models in the search space and train them for up to 3 hours or until the validation performance fails to increase for more than 128 epochs. The best model among the models sampled chosen according to validation performance obtained among the 16 sampled models has test accuracy equal to 99.72%, which is close to the single model state-of-the-art of 99.77% (Sato et al., 2015). Additionally, taking a simple majority voting emsemble of the 5 best performing models yielded the same validation accuracy as the best single model and increased test accuracy to 99.75%. The performance profile of the sampled models and the architecture and hyperparameters of the best model are presented in Appendix A.

We can build good ensembles by sampling models in the search space and building an ensemble out of the best ones. It has been observed in the literature that model diversity often improves emsemble performance. Our results suggest that it is possible to define search spaces that work well across a range of tasks, having the potential to significantly reduce the burden on the human expert.

## 8 CONCLUSION

We described a framework for automatically designing and training deep models. This framework consists of three fundamental components: the model search space specification language, the model search algorithm, and the model evaluation algorithm. The model search space specification language is composable, modular, and extensible, and allows us to easily define expressive search spaces over architectures. The model evaluation algorithm determines how to compute a score for a model in the search space. Models can be automatically compiled to their corresponding computational graphs. Using the model search space specification language and the model evaluation algorithm, we can introduce model search algorithms for exploring the search space. Using our framework, it is possible to do random search over interesting spaces of architectures without much effort from the expert. We also described more complex model search algorithms, such as MCTS, MCTS with tree restructuring, and SMBO. We present experiments on CIFAR-10 comparing different model search algorithms and show that MCTS with tree restructuring and SMBO outperform random search. Code for our framework and experiments has been made publicly available. We hope that this paper will lead to more work and better tools for automatic architecture search.

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

## A  DETAILED EXPERIMENTAL SETUP

In Section 7, we considered a search space of deep convolutional models having structural hyperparameters for the depth of the network, whether to apply batch normalization before or after ReLU, and whether to use dropout; hyperparameters for the number and size of the convolutional filters; training hyperparameters for the learning rate schedule. We show in Figure 4 the LISP-like pseudocode for the search space considered in Section 7, and in Figure 5 the corresponding runnable Python implementation in our framework.

```
MH = (UserHyperparams
        { 'optimizer_type' : ['adam', 'sgd'],
          'learning_rate_init' : logspace(10^-2, 10^-7, 32),
          'rate_mult' : logspace(10-2, 0.9, 8),
          'rate_patience' : [4, 8, 12, 16, 20, 24, 28, 32],
          'stop_patience' : [64],
          'learning_rate_min' : [10^-9] })

M1 = (Conv2D [48, 64, 80, 96, 112, 128] [3, 5, 7] [2])

M2 = (RepeatTied
        (Concat
            (Conv2D [48, 64, 80, 96, 112, 128] [3, 5] [1])
            (MaybeSwap BatchNormalization ReLU)
            (Optional (Dropout [0.5, 0.9])))
        [1, 2, 4, 8, 16, 32])

M = (Concat MH M1 M2 M1 M2 (Affine [10]))
```

Figure 4: Specification of the model search space used in Section 7 in LISP-like pseudocode. See Figure 5 for the corresponding runnable Python code.

In Figure 4 and Figure 5, to include training hyperparameters in the search space, we concatenate the module that encapsulates the training hyperparameters (the module assigned to MH) and the modules that encapsulate the remaining model hyperparameters (the modules other than MH in the declaration of M).

The Python specification of the model search space in Figure 5 is remarkably close in both semantics and length to the LISP-like pseudocode in Figure 4. We omit some hyperparameters in Figure 4 because we did not consider multiple values for them, e.g., for Conv2D modules, we always used same size padding and the initialization scheme described in He et al. (2015).

Our implementation has code modularity and reusability benefits. For example, we can define an auxiliary function to instantiate modules and then use it in the instantiation of the module for the complete search space. This is illustrated in Figure 5 with the definition of Module_fn and its use in the declaration of M.

See Figure 6a for the performance profile of 16 models randomly sampled from the search space in Figure 5. See Figure 6b for the architecture and training hyperparameters of the best model found in the 16 samples.

## B  LIST OF MODULES

We provide a brief description of a representative subset of the types of basic and composite modules that we have implemented in our framework. It is simple to define new modules this list by implementing the module interface described in Section C.

### B.1  BASIC MODULES

Basic modules take no other modules when instantiated, having only local hyperparameters and parameters.

- Affine: Dense affine transformation. Hyperparameters: number of the hidden units and initialization scheme of the parameters. Parameters: dense matrix and bias vector.

```
MH = UserHyperparams(['optimizer_type',
                      'learning_rate_init',
                      'rate_mult',
                      'rate_patience',
                      'stop_patience',
                      'learning_rate_min',
                      'angle_delta',
                      'scale_delta',
                      'weight_decay_coeff'],
                     [['adam', 'sgd_mom'],
                      list( np.logspace(-2, -6, num=32) ),
                      list( np.logspace(-2, np.log10(0.9), num=8) ),
                      range(8, 65, 4),
                      [128],
                      [1e-6],
                      [0, 5, 10, 15, 20, 25, 30, 35],
                      [0.0, 0.05, 0.1, 0.15, 0.2, 0.25, 0.3, 0.35],
                      [0.0, 1e-6, 1e-5, 1e-4] ])

conv_initers = [ kaiming2015delving_initializer_conv(1.0) ]
aff_initers = [ xavier_initializer_affine( 1.0 )]

def Module_fn(filter_ns, filter_ls, keep_ps, repeat_ns):
    b = RepeatTied(
            Concat([
                Conv2D(filter_ns, filter_ls, [1], ["SAME"], conv_initers),
                MaybeSwap_fn( ReLU(), BatchNormalization() ),
                Optional_fn( Dropout(keep_ps) )
        ]), repeat_ns)
    return b

filter_nums = range(48, 129, 16)
repeat_nums = [2 ** i for i in xrange(6)]
mult_fn = lambda ls, alpha: list(alpha * np.array(ls))

M = Concat([MH,
            Conv2D(filter_nums, [3, 5, 7], [2], ["SAME"], conv_initers),
            Module_fn(filter_nums, [3, 5], [0.5, 0.9], repeat_nums),
            Conv2D(filter_nums, [3, 5, 7], [2], ["SAME"], conv_initers),
            Module_fn(mult_fn(filter_nums, 2), [3, 5], [0.5, 0.9], repeat_nums),
            Affine([num_classes], aff_initers) ])
```

Figure 5: Runnable specification of the model search space used in Section 7 in our Python implementation of the framework. See Figure 4 for the specification of the same search space in the LISP-like pseudocode used throughout this paper.

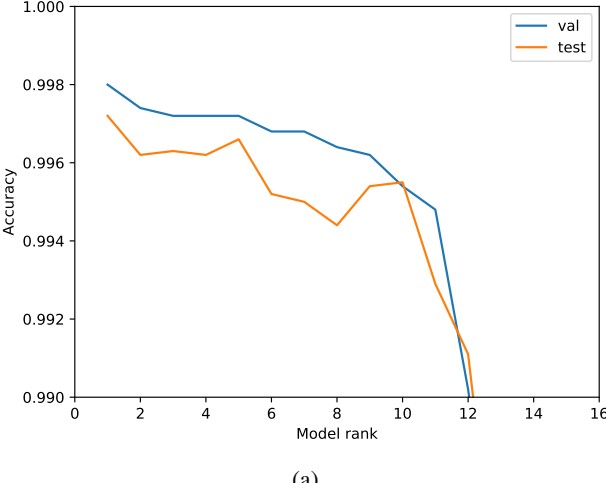

(a)

```
( ( 'UserHyperparams',
'adam',
0.003046989570903508,
0.24882127247602889,
52,
128,
1e-06,
15,
0.1,
1e-06),
('Conv2D', 80, 7, 2, 'SAME'),
('Conv2D', 96, 3, 1, 'SAME'),
('ReLU',),
('BatchNormalization',),
('Dropout', 0.9),
('Conv2D', 96, 3, 1, 'SAME'),
('ReLU',),
('BatchNormalization',),
('Dropout', 0.9),
('Conv2D', 96, 3, 1, 'SAME'),
('ReLU',),
('BatchNormalization',),
('Dropout', 0.9),
('Conv2D', 96, 3, 1, 'SAME'),
('ReLU',),
('BatchNormalization',),
('Dropout', 0.9),
('Conv2D', 96, 3, 1, 'SAME'),
('ReLU',),
('BatchNormalization',),
('Dropout', 0.9),
('Conv2D', 96, 3, 1, 'SAME'),
('ReLU',),
('BatchNormalization',),
('Dropout', 0.9),
('Conv2D', 96, 3, 1, 'SAME'),
('ReLU',),
('BatchNormalization',),
('Dropout', 0.9),
('Conv2D', 96, 3, 1, 'SAME'),
('ReLU',),
('BatchNormalization',),
('Dropout', 0.9),
('Conv2D', 96, 3, 1, 'SAME'),
('ReLU',),
('BatchNormalization',),
('Dropout', 0.9),
('Conv2D', 128, 7, 2, 'SAME'),
('Conv2D', 128, 3, 1, 'SAME'),
('BatchNormalization',),
('ReLU',),
('Dropout', 0.5),
('Affine', 10))
```

(b)

Figure 6: (a) The performance profile of the 16 sampled models in decreasing order of their validation accuracy. The model with the highest validation accuracy (99.80%) has also the highest test accuracy (99.72%). (b) The best performing model found from sampling 16 models of search space in Figure 5 with a random model searcher. The hyperparameters of `UserHyperparams` are as in the search space in Figure 5. The hyperparameters of the layers are as described in Appendix B. The parameters of the `Affine` and `Conv2d` modules were initialized according to Glorot & Bengio (2010) and He et al. (2015), respectively.

- `ReLU`: ReLU nonlinearity. Hyperparameters: none. Parameters: none.

- `Dropout`: Dropout. Hyperparameter: dropout probability. Parameters: none.

- `Conv2D`: Two-dimensional convolution. Hyperparameters: number of filters, size of the filters, stride, padding scheme, and initialization scheme of the parameters. Parameters: convolutional filters and bias vector.

- `MaxPooling2D`: Two-dimensional max pooling. Hyperparameters: size of the filters, stride, and padding scheme. Parameters: none.

- `BatchNormalization`: Batch normalization. Hyperparameters: none. Parameters: translation coefficients and scaling coefficients.

- `UserHyperparams`: User-defined hyperparameters. Hyperparameters: hyperparameters determined by the user expert. Parameters: none.

- `Empty`: Identity. Hyperparameters: none. Parameters: none.

## B.2 COMPOSITE MODULES

Composite modules take other modules as arguments when instantiated, which we will call *submodules*. The behavior of a composite module depends on its submodules. The hyperparameters which a composite module has to specify depend on the values of the hyperparameters of the composite module and the hyperparameters of the submodules; e.g., `Or` takes a list of submodules but it only has to specify the hyperparameters of the submodule that it ends up choosing. A composite module is responsible for specifying its submodules, which is done through calls to the module interfaces of the submodules.

- `Concat`: Takes a list of submodules and connects them in series. Hyperparameters: hyperparameters of the submodules. Parameters: parameters of the submodules.

- `Or`: Chooses one of its submodules to use. Hyperparameters: which submodule to use and hyperparameters of the submodule chosen. Parameters: parameters of the submodule chosen.

- `Repeat`: Repeats a submodule some number of times, connecting the repetitions in series; values for the hyperparameters of the repetitions are chosen independently. Hyperparameters: number of times to repeat the submodule and hyperparameters of the repetitions of the submodule. Parameters: parameters of the repetitions of the submodule.

- `RepeatTied`: Same as `Repeat`, but values for the hyperparameters of the submodule are chosen once and used for all the submodule repetitions. Hyperparameters: the number of times to repeat the submodule and hyperparameters of the submodule. Parameters: parameters of the repetitions of the submodule.

- `Optional`: Takes a submodule and chooses whether to use it or not. Hyperparameters: whether to include the submodule or not and, if included, hyperparameters of the submodule. Parameters: if included, parameters of the submodule.

- `Residual`: Takes a submodule and implements a skip connection adding the input and output; if the input and output have different dimensions, they are padded to make addition possible. Hyperparameters: hyperparameters of the submodule. Parameters: parameters of the submodule.

- `MaybeSwap`: Takes two submodules and connects them in series, choosing which submodule comes first. Hyperparameters: which of the submodules comes first and hyperparameters of the submodules. Parameters: parameters of the submodules.

## C MODULE INTERFACE

We describe the module interface as we implemented it in Python. To implement a new type of module, one only needs to implement the module interface.

```
class Module(object):
    def initialize(self, in_d, scope)
    def get_outdim(self)
    def is_specified(self)
    def get_choices(self)
    def choose(self, choice_i)
    def compile(self, in_x, train_feed, eval_feed)
```

Figure 7: Module interface used by all modules irrespective if they are basic or composite. To implement a new type of module, the human expert only needs to implement the module interface.

- `initialize`: Tells a module its input dimensionality. A composite module is responsible for initializing the submodules that it uses.

- `get_outdim`: Once a module is fully specified, we can determine its output dimensionality by calling `get_outdim`. The output dimensionality is a function of the input dimensionality (which is determined when `initialize` is called) and the values of the hyperparameters chosen.

- `is_specified`: Tests whether a module is fully specified. If a module is fully specified, `outdim` and `compile` may be called.

- `get_choices`: Returns a list of the possible values for the hyperparameter currently being specified.

- `choose`: Chooses one of the possible values for the hyperparameter being specified. The module assigns the chosen value to that hyperparameter and either transitions to the next hyperparameter to specify or becomes fully specified. The module maintains internally the state of its search process.

- `compile`: Creates the computational graph of the model in a deep learning model specification language, such as Tensorflow or PyTorch. For composite modules, compilation can be performed recursively, through calls to the `compile` functions of its submodules.

Composite modules rely on calls to the module interfaces of its submodules to implement their own module interfaces. For example, `Concat` needs to call `out_dim` for the last submodule of the series connection to determine its own output dimensionality, and needs to call `choose` on the submodules to specify itself. One of the design choices that make the language modular is the fact that a *composite module can implement its own module interface through calls to the module interfaces of its submodules*. All information about the specification of a module is local to itself or kept within its submodules.

## D  BEYOND SINGLE-INPUT SINGLE-OUTPUT MODULES

We can define new modules with complex signal paths as long as their existence is encapsulated, i.e., a module may have many signal paths as long they fork from a single input and merge to a single output, as illustrated in Figure 8.

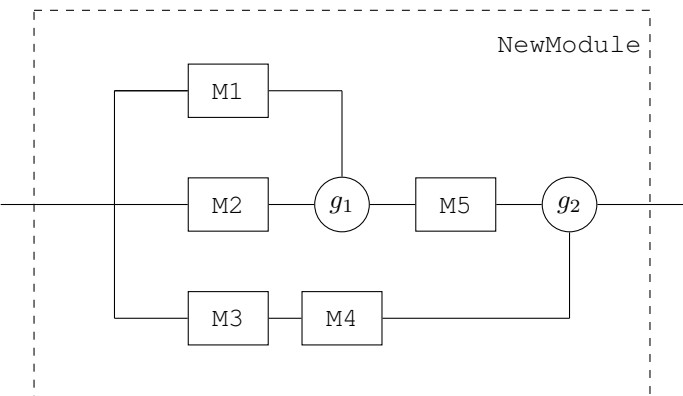

Figure 8: A module with many signal paths from input to output. To implement a module, the human expert only needs to implement its module interface. `M1`, `M2`, `M3`, and `M4` are arbitrary single-input single-output modules; $g_1$ and $g_2$ are arbitrary transformations that may have additional hyperparameters. The hyperparameters of $g_1$ and $g_2$ can be managed internally by `NewModule`.

In Figure 8 there is a single input fed into `M1`, `M2`, and `M3`. `M1`, `M2`, `M3`, `M4`, `M5` are arbitrary single-input single-output submodules of `NewModule`. The module interface of `NewModule` can be implemented using the module interfaces of its submodules. Instantiating a module of type `NewModule` requires submodules for `M1`, `M2`, `M3`, `M4`, and `M5`, and potentially lists of possible values for the hyperparameters of $g_1$ and $g_2$. A residual module which chooses what type of merging function to apply, e.g., additive or multiplicative, is an example of a module with hyperparameters for the merging functions

A module of the type `NewModule` is fully specified after we choose values for all the hyperparameters of `M1`, `M2`, `M3`, `M4`, `M5`, $g_1$, and $g_2$. Testing if `M1`, `M2`, `M3`, `M4`, and `M5` are fully specified can be done by calling `is_specified` on the corresponding submodule.

The output dimensionality of `NewModule` can be computed as a function of the values of the hyperparameters of $g_2$ and the output dimensionality of `M5` and `M4`, which can be obtained by calling `get_outdim`. Similarly, for `get_choices` we have to keep track of which hyperparameter we are specifying, which can either come from `M1`, `M2`, `M3`, `M4`, and `M5`, or from $g_1$ and $g_2$. If we are choosing values for an hyperparameter in `M1`, `M2`, `M3`, `M4`, and `M5` we can call `get_choices` and `choose` on that submodule, while for the hyperparameters of $g_1$ and $g_2$ we have to keep track of the state in `NewModule`. `compile` is similar in the sense that it is implemented using calls to the `compile` functionality of the submodules.

