# OpenReview forum: "DeepArchitect: Automatically Designing and Training Deep Architectures"
_ICLR.cc/2018/Conference — Reject_

### Official Review · AnonReviewer1 · 2017-11-21
**The novelty in this paper is below what is expected for a publication at ICLR. I recommend rejection.**

**Rating:** 4
**Confidence:** 5

**Review:**

The author present a language for expressing hyperparameters (HP) of a network. This language allows to define a tree structure search space to cover the case where some HP variable exists only if some previous HP variable took some specific value. Using this tool, they explore the depth of the network, when to apply batch-normalization, when to apply dropout and some optimization variables. They compare the search performance of random search, monte carlo tree search and a basic implementation of a Sequential Model Based Search.

The novelty in this paper is below what is expected for a publication at ICLR. I recommend rejection.

---

> ### Author Response · Authors · 2018-01-05
> **The reviewer misrepresents our work**
>
> Our work is novel for the reasons elaborated in our other replies.
>
> Since this work has been made available on ArXiv, its ideas have been used or suggested future work on architecture search. For example, the ideas of compositionality for the representation of a search space are used in Hierarchical Representations for Efficient Architecture Search (https://openreview.net/forum?id=BJQRKzbA-); using MCTS to do model search is used in "Finding Competitive Network Architectures Within a Day Using UCT"; using SMBO to do model search is used in "Progressive Neural Architecture Search", among others.
>
> The framework set forth by this paper provides a foundation for thinking about architecture search. Progress in architecture search can be made by developing a better model search space representation language, giving more expressive tools to a deep learning expert to represent search spaces over models. Progress can also be made by developing better model search algorithms that search spaces of models more efficiently. The fact that this framework is modular is a big advantage, as research can be focused on each of the components rather that having to having to develop a monolithic system from scratch each time.

---

### Official Review · AnonReviewer3 · 2017-11-24
**The authors propose to automatically design and train deep architectures.**

**Rating:** 5
**Confidence:** 3

**Review:**

This paper introduces a DeepArchitect framework to build and train deep models automatically. Specifically, the authors proposes three components, i.e., model search specification language, model search algorithm, model evaluation algorithm. The paper is written well, and the proposed framework provides us with a systematical way to design deep models.

However, my concern is mainly about its importance in practice. The experiments and computational modules are basic and small-scale, i.e., it may be restricted for large-scale computer vision problems.

---

> ### Author Response · Authors · 2018-01-05
> **The decomposition of the framework in the three parts: model search space specification language, model search algorithm, and model evaluation algorithm allows us to reason clearly about architecture search. The model search space specification is modular, compositional, and extensible.**
>
> The reviewer presents the following criticism:
> 1. The decomposition of the framework into model search space specification language, model search algorithm, and model evaluation algorithm is interesting, but there is concern about its importance in practice.
> 2. There is concern that the framework may be restrictive for large-scale problems.
>
> Response:
> 1. The decomposition into these three components allows us to think clearly about each of them rather than dealing with all aspects simultaneously. For example, future contributions may focus on extending the model search space specification language or on developing better model search algorithms. These components interact only through a very minimal interface. This decomposition of the problem will be useful for future research on architecture search.
>
> 2. There is no fundamental reason why this framework should be restrictive in the way the reviewer is concerned. We discuss in Section 4 the properties required by a module. These are quite general, and therefore we can easily introduce useful new basic and composite modules. See also Appendix B for examples of modules that we defined. As the different components in our framework are highly extensible and modular, our work will be very useful in approaching future problems in architecture search. Namely, the model search space specification language is expressive enough to capture many relevant high-performance search spaces, as discussed in paragraph 4 of section 4.1 and in appendix D.
>
> We do not have the resources available to conduct experiments on the scale of some other recent papers (e.g., those coming out of major corporate research labs such as Google). Nonetheless, our smaller scale experiments are enough to support our claims: expressive search spaces over architectures can be represented easily by writing expressions in our model search space specification language (see Appendix A, Figure 4, and Figure 5); the search spaces induced can be effectively searched by random search; using model search algorithms attuned to the structure of the search space results in improved search performance.
>
> One of the main focus of this work is the representation power of the model search space specification language. We also note that due to the flexibility of the search space specification language, architecture search can be easily integrated in a ML workflow, as the expert only has to design a reasonable search space and provide a way of evaluating models.

---

### Official Review · AnonReviewer2 · 2017-11-29
**MCTS is promising, but should be evaluated in a more standard way**

**Rating:** 4
**Confidence:** 5

**Review:**

Monte-Carlo Tree Search is a reasonable and promising approach to hyperparameter optimization or algorithm configuration in search spaces that involve conditional structure.

This paper must acknowledge more explicitly that it is not the first to take a graph-search approach. The cited work related to SMAC and Hyperopt / TPE addresses this problem similarly. The technique of separating a description language from the optimization algorithm is also used in both of these projects / lines of research. The [mis-cited] paper titled “Making a science of model search …” is about using TPE to configure 1, 2, and 3 layer convnets for several datasets, including CIFAR-10. SMAC and Hyperopt have been used to search large search spaces involving pre-processing and classification algorithms (e.g. auto-sklearn, autoweka, hyperopt-sklearn). There have been near-annual workshops on AutoML and Bayesian optimization at NIPS and ICML (see e.g. automl.org).

There is a benchmark suite of hyperparameter optimization problems that would be a better way to evaluate MCTS as a hyperparameter optimization algorithm: http://www.ml4aad.org/automl/hpolib/

---

> ### Author Response · Authors · 2018-01-05
> **Our goal is to develop a framework for architecture search in deep learning around the model search space specification language introduced. Our goal is not to propose MCTS as an algorithm for general purpose hyperparameter optimization; rather we propose random, MCTS, and SMBO as simple baseline algorithms for model search that are well-suited to the structures arising from the search spaces induced by our model search space specification language.**
>
> The reviewer presents the following criticism:
> 1. TPE/SMAC also allows us to describe search spaces with conditional structure.
> 2. “Making a science of model search …” uses TPE to search over simple convolutional architectures.
> 3. The performance of MCTS for hyperparameter optimization would be better evaluated in the benchmarks pointed out by the reviewer.
>
> Response:
> 1. We do not claim to be the first to present a method that works with conditional structure. We clearly state in our paper that there are general-purpose hyperparameter optimization algorithms such as TPE, however these are harder to use for architecture search because they require the user to write more code and single out what are the hyperparameters to search over. In contrast, in our, writing an expression in our DSL (domain-specific language) automatically induces the search space. Furthermore, this language allows us to directly compile the resulting model to the corresponding computational graph. Note that our work in focused on architecture search for deep learning and not general hyperparameter optimization. We are in the same line of work as Zoph and Le (ICLR 2017).
>
> Expressions written in our model search space specification language encode trees; paths through the encoding correspond to fully specified models that can be compiled to computational graphs. Note that this tree is implicit;  we only require functionality to traverse the tree, and not a full explicit representation. This is important when there are exponentially many paths from the root to leaves. This contrasts with TPE: e.g., "Making a science of model search" uses a simple representation, and therefore is constrained to simple trees. For example, the following (toy) search space is hard to represent in Hyperopt, but poses no problem in our language:
>
> (Repeat
>   (Optional
>     (Repeat
>       (Affine [32, 64])
>      [1, 2, 4])
>   )
> [1, 2, 4])
>
> The problem arises from the interaction between Optional and Repeat. These problems are exacerbated by deeper nesting. Representing this search space in Hyperopt would require writing a cumbersome cases expression for each of the different combinations of hyperparameter values for Repeat and Optional modules. See https://github.com/jaberg/hyperopt/wiki/FMin for information on the cases construct in Hyperopt: hp.choice. By contrast, our language imposes no such burden on the user.
>
> Due to these significant differences, it is incorrect to say that our DSL for specifying search spaces over architectures is not novel when compared to something such as Hyperopt.
>
> We explore how the introduction of the search space specification language allows us to construct a integrated framework for architecture search. The main focus of this paper is not to propose new hyperparameter optimization algorithms in current general settings. The model search algorithms that we propose are adjusted to the structures that arise from our model search space specification language. The experiments study the potential of different model search algorithms with structures of this type.
>
> 2. The paper mentioned does indeed search over simple convolutional architectures on CIFAR-10, nonetheless, the search space is hard-coded and does not make use of the compositionality resulting from the model search space specification language. This point is addressed in the related work section (e.g., paragraph 6 of section 2). One important aspect of our framework is that it allows the user to easily write search spaces over architectures, functioning as a tool to support model discovery. Much of the recent progress in deep learning was supported by the existence of tools that allow experts rapid experimentation and exploration (e.g., Tensorflow, Pytorch). We need to build these tools for architecture search in deep learning, i.e., specific tools for architecture search rather than existing general-purpose hyperparameter optimization tools.
>
> 3. Our work focus on the development of a framework for architecture search in deep learning. Currently, there are no standard benchmarks for architecture search. In particular, due to the focus on architecture search, the suggested generic datasets for pipeline tuning (e.g., auto-sklearn, autoweka) would fit well the message of this paper (e.g., from paragraph 3 of section 1 to the end of that section). Our goal is not to propose MCTS as an algorithm for general purpose hyperparameter optimization; rather we propose random, MCTS, and SMBO as simple baseline algorithms for model search that are well-suited to the structures arising from the search spaces induced by our model search space specification language.

---

> > ### Author Response · Authors · 2018-01-05
> > **Final remarks on the previous reply**
> >
> > Final remarks:
> > The introduction of the model search space specification language along with just random search experiments would by itself be interesting enough to warrant publication. This domain-specific language is extensible and compositional, allowing the user to easily represent search spaces over architectures and compile them to computational graphs. The model search algorithms proposed, while simple, are well-suited to the resulting search spaces and are a good start to design more complex and performant ones for this setting. This is, to the best of our knowledge, the first work to propose such a DSL for architecture search and explore its benefits; as such, it provides an extensible platform for future research on architecture search. To support this benefit, we have made all code available and will continue to extend it.

---

> > > ### Comment · AnonReviewer2 · 2018-01-11
> > > **re: final remarks**
> > >
> > > A domain specific language would be more appreciated by a different audience, where writing in Python is seen as a drawback and where the DSL is more exciting. As a reviewer for ICLR I am looking for something new that it lets me do. I don't see it yet, and random search wouldn't be compelling.
> > >
> > > Also, there has been some mis-characterization of Hyperopt here: Hyperopt has always supported graph-structured (rather than just tree-structured) compositions in the computational graph / search space input to fmin. The "Tree of Parzen Windows" is something of a misnomer in this regard, as the implementation of that algorithm in Hyperopt works on directed acyclic graphs.

---

> > > > ### Author Response · Authors · 2018-01-12
> > > > **re: re: final remarks**
> > > >
> > > > Saying that a DSL is not interesting for this community is your subjective opinion. Work on code, frameworks, and APIs has been published at top ML conferences and/or been hugely influential in ML; for example:
> > > > - http://download.tensorflow.org/paper/whitepaper2015.pdf
> > > > - https://papers.nips.cc/paper/5872-efficient-and-robust-automated-machine-learning.pdf
> > > > - https://papers.nips.cc/paper/6986-on-the-fly-operation-batching-in-dynamic-computation-graphs.pdf,
> > > > - https://openreview.net/forum?id=ryrGawqex
> > > > - https://arxiv.org/pdf/1701.03980.pdf
> > > > - https://papers.nips.cc/paper/6256-a-credit-assignment-compiler-for-joint-prediction.pdf).
> > > >
> > > > You can always take the position that something is not interesting because it does not let you do anything new. Using that line of thought, high-level programming languages are not interesting because you already can accomplish the same in low-level programming languages; Tensorflow, PyTorch, and similar framework are not interesting because you can write your own deep learning code from scratch in C++ or Python.
> > > >
> > > > Much of the recent progress in ML was greatly facilitated by the existence of high-level tools such Tensorflow and Pytorch. One can only wonder how much more far behind as a community we would be if everyone was still writing their own backpropagation implementations. While Tensorflow or Pytorch do not allow you to do things that could not be done in C++ or Python, they make expressing interesting ML programs drastically easier, and as a result, researchers are able to think about and approach problems differently than they would if there were no such deep learning frameworks.
> > > >
> > > > Our work allows us to think about architecture search differently. Namely, splitting the problem of architecture search into three parts (model search space specification language, model search algorithm, model evaluation algorithm) is a useful perspective as the different parts can be changed and/or improved independently (e.g., a more expressive model search space specification language, a more sample efficient model search algorithm, a different model evaluation algorithm). Additionally, our model search space specification language has interesting ideas on how to induce these complex compositional architecture search spaces (which end up being in complex implicit hyperparameter spaces) by writing expressions in a DSL. The fact that all this could be expressed differently is of limited relevance. The truth is that our DSL allows researchers to think differently about the problem and gives them interesting tools to write expressive search spaces over architectures. That being said, we believe that our paper introduces a sufficient number of new and interesting ideas to be of value to this community.
> > > >
> > > > Can you clarify (e.g., give an example) of what you mean by "directed acyclic graph" in this context (i.e., in the context of hyperparameters spaces).

---

### Decision · Program_Chairs · 2018-01-29
**ICLR 2018 Conference Acceptance Decision**

**Decision:**

Reject

**Comment:**

This paper introduces a framework for specifying the model search space for exploring over the space of architectures and hyperparameters in deep learning models (often referred to as architecture search).  Optimizing over complex architectures is a challenging problem that has received significant attention as deep learning models become more exotic and complex.  This work helps to develop a methodology for describing and exploring the complex space of architectures, which is a challenging problem.  The authors demonstrate that their method helps to structure the search over hyperparameters using sequential model based optimization and Monte Carlo tree search.

The paper is well written and easy to follow.  However, the level of technical innovation is low and the experiments don't really demonstrate the merits of the method over existing strategies.  One reviewer took issue with the treatment of related work.  The underlying idea is compelling and addresses an open question that is of great interest currently.  However, without experiments demonstrating that this works better than, e.g., the specification in the hyperopt package, it is difficult to assess the contribution.  The authors must do a better job of placing this contributing in the context of existing literature and empirically demonstrate its advantages.  The presented experiments show that the method works in a limited setting and don't explore optimization over complex spaces (i.e. over architectures - e.g. number of layers, regularization for each layer, type of each layer, etc.).  There's nothing presented empirically that hasn't been possible with standard Bayesian optimization techniques.

This is a great start, but it needs more justification empirically (or theoretically).

Pros:
- Addresses an important and pertinent problem - architecture search for deep learning
- Provides an intuitive and interesting solution to specifying the architecture search problem
- Well written and clear

Cons:
- The empirical analysis does not demonstrate the advantages of this approach over existing literature
- Needs to place itself better in the context of existing literature